# Proposal of a Holistic Framework to Support Sustainability of New Product Innovation Processes

**Ana S. M. E. Dias** [1,2,*], **António Abreu** [1,3], **Helena V. G. Navas** [2] **and Ricardo Santos** [1,4]

1    ADEM-ISEL, Departmental Area of Mechanical Engineering of Superior Institute of Engineering of Lisbon, Polytechnic Institute of Lisbon, 1959-007 Lisbon, Portugal; ajfa@dem.isel.ipl.pt (A.A.); ricardosimoessantos84@ua.pt (R.S.)
2    UNIDEMI, Department of Mechanical and Industrial Engineering, NOVA School of Science and Technology, Universidade NOVA de Lisboa, 2829-516 Caparica, Portugal; hvgn@fct.unl.pt
3    CTS, UNINOVA, NOVA School of Science and Technology, Universidade NOVA de Lisboa, 2829-516 Caparica, Portugal
4    GOVCOPP—University of Aveiro, 3810-193 Aveiro, Portugal
*    Correspondence: ana.dias@isel.pt

**Abstract:** The survival of companies in globalized and highly competitive markets, heavily depends on their ability to innovate through the creation of new products and/or services, supported by sustainable processes to prevent business failure. There are many factors regarding the interface company/stakeholders/market at all hierarchical levels, which have a major contribution to sustain innovation in processes regarding the creation of new products and services. A holistic approach of all these factors, as a whole, has not been a subject of scientific research conducting to the necessity of creating a proposal of a framework that can be integrated and systemic. Thus, this paper aims to propose a functional holistic model, which integrates the strategic, organizational and operational levels regarding market business and company interaction, as well as the set of factors to take into account to guarantee assurance that innovative processes are sustained, when new products and/or services are created or improved. Conducted through an investigation of the state of the art, by literature review, a comprehensive and integrated conceptual model was built in a deductive-inductive way. Then, the conceptual model was validated through four case studies. Finally, it was found that the conceptual framework became functional, because its applicability has been successfully tested in a business environment. As a result, the tool developed here, can be useful to measure and evaluate projects dedicated to companies that innovate in a sustainable way.

**Keywords:** sustainability; innovation; new products; functional framework; SIFSNPIP; case studies

## 1. Introduction

Competitive new products and services are the output of sustainable innovative processes that companies manage in a systematic way, challenged by demanding and dynamic business markets [1]. According to [2], sustainable innovative processes are crucial for the survival of competitive companies, being a major factor for business success [3]. Also, firms should undertake their ideas about sustainable innovative products and services, and bring it to market as quickly that they can, to be competitive in nowadays global markets [4]. But the sustainability of innovative processes that support new product creation is not an easy process, and therefore a project can fail even when it was initially estimated to succeed. So, the innovative processes developed to create new sustainable products, involves considerable and various risks due to the uncertainty associated to business markets, according to [5]. Thus, risk is a strong obstacle to be transposed in business market characterizes by uncertainty,

complexity and turbulence, according to [2,6]. So, the shortening of the available time to manage projects with both efficiently and effectively, is a very important issue to be taken in account by managers, especially when they concern relating to products and services of radical innovative nature [7]. According to [6], companies should manage their projects in a proactive, structured and sustainable way, to survive and succeed in such competitive markets, and for that, sharing knowledge through collaborative networks is crucial [8], taking in account all possible variables and parameters that have influence in the strategic, organizational and operational hierarchical levels of companies [9]. This finding is extremely important regarding new products and services that emerge of innovative projects and their sustainable implementation, which requires an increasing rational and holistic approach [10]. It is extremely challenging for companies the development of successful sustainable innovative processes to create new products or services, and for that the path that involves generation of new ideas is crucial. New idea generation occurs normally in the beginning of sustainable innovation processes to generate new products and services. So, this point is especially important, since it determines companies potential to undertake promising new product and service ideas at reasonable costs. In contexts where resources are constrained, creativity seems to be extremely contributive to problem-solving processes [11].

During literature review, was found a lack of holistic approaches or frameworks that could encompass the strategic, organizational and operational hierarchical levels of companies, in order to create new products and services through sustainable innovative projects, aiming the minimization of the risks of business failure inherent to its implementation. To serve these needs, it's proposed in this paper a "Systemic and Integrated Framework for Sustainability of New Product Innovation Processes"—SIFSNPIP. Thus, an extensive literature review supporting the construction of conceptual version of SIFSNPIP was carried out and it's presented on Section 2 of this paper. In Section 3, the key phases regarding the research methodology approached in this paper are presented. Section 4 presents and describes the case studies that were carried out to validate conceptual SIFSNPIP model and shows the aspects that were found which allowed to transform the model from conceptual to functional. In Section 5 the full framework of SIFSNPIP it's presented. Finally, the main of this investigation conclusions are presented in Section 6.

## 2. Literature Review

In order to design a theoretical framework, that determines the sustainability of innovative processes in the creation of new products or services, it was needed to find the most relevant set of variables and parameters that comprise strategic, corporative and operational business levels, as well as the way they interact with each other.

Firstly the approach of the strategic level was needed in order to understand which aspects embrace the company/market interactions and its ways of articulation. Secondly was found that the organizational level can be decomposed into two sub-levels with the same importance to a company: the corporate culture with a structural nature and the management principles that normally respond with market situations. Thirdly, the operational level was approached, as well as its inherent processual variables. So, the developed literature review was organized in this sequence, as Figure 1 illustrates. In there, the arrows between all levels, show the way of the relationship that the three levels have among themselves.

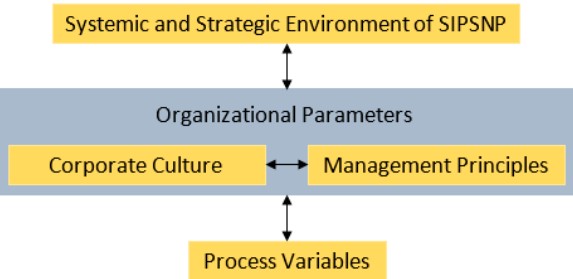

**Figure 1.** General framework of the approach of the Systemic and Integrated Framework for Sustainability of New Product Innovation Processes (SIFSNPIP) (authors' own elaboration).

*2.1. Systemic and Strategic Environment in Sustainability of Innovative Processes that Support New Products (SIPSNP)*

In the strategic level of the SIPSNP, is crucial for managers to take in account the relationship between: sustainable strategic innovation; source of new product or service creation; and the concept of business strategy. Sustainable strategic innovation in a highly competitive environment, embodies industries regarding both disruptive and incremental nature. The first one exists in markets without competitors, designated by "blue ocean strategy" (BOS), and the second exists whenever there is competition, designated by "red ocean strategy" (ROS) [12].

BOS is especially important when companies and businesses need to grow fast, and rather than compete with existing rivals as happens in ROS, BOS allows the creation of unique offerings for the emerging new markets [12].

Despite the above mentioned, is normal that hybrid strategies to be very common in business markets in which firms develop sustainable new products and services that emanate from innovation that embraces both radical and gradual processes, and in this context, hybrid strategies are called "purple ocean strategy" (POS) [13]. POS strategy corresponds to disruptive products and services with no competition at the beginning and while the possibility to remain like that exists. At the same time, other products regarding incremental innovation faces the existent competition. Beyond these strategies, companies have more challenges involving other factors of strategic nature that must be also considered, thus they require accurate analysis of correspondent trade-offs involved, namely facts concerning various risks and their interaction [14]. In a competitive environment along with high complexity of production processes risks of business failure must be analyzed in a systematic way, which means, with a continuous analysis of trade-offs involving the various risk factors of SIPSNP, especially those regarding quality, time and costs [15].

Independently of the SIPSNP strategies being radical, incremental or mixed, companies can't ignore the risks which they are exposed and must to be aware at the dynamics of competition through the implementation of systematic benchmarking practices [16]. For companies achieve the best performances of SIPSNP business, [17] concluded that teamwork, multidisciplinary and collaborative attitude have a huge impact in benchmarking practices efficiency and effectiveness, which should integrate the corporate culture of firms. According to [18], the globalization of markets and businesses is a trend that will remain strong in long term. Therefore an unavoidable aspect of globalization has been outsourcing practices, especially the knowledge-based services, such as the development of SIPSNP. Companies all over the world need to reduce costs, but the question is not the need of a particular practice of outsourcing or working abroad, but when and how it will be done to achieve greater competitive advantage in the market [18]. One special fact that arises in a particular process of outsourcing and/or offshoring is the "intellectual property" (IP) jointly developed. According to [18], exploitation and defense of IP, when generating both incremental and radical innovation, have impact on the strategic management of the focal company. Globalization and internationalization of business regarding SIPSNP projects, often correspond to engineering and management complex systems involving R&D and information highly reserved [19]. The strategic options pointed out are

associated to risks of opportunistic expropriation of knowledge and related monitoring costs of the subcontracted partners, which sometimes are not only distant in geography but also in culture [19]. The focal company must have enough responsibility, knowledge and skills to ensure control over the processes regarding, onshore/offshore, third-party option (third-party logistics 3pl) decisions to guarantee that the final product or service fulfils the customer's needs. Therefore, all stakeholders involved in a business must integrate a network of synergies, to ensure the articulation of all processes regarding SIPSNP projects, but this fact increases complexity to the whole system and the concomitant risk that can emerge from the failure of each element. That's why the risks must be predicted so they can be avoided when trade-offs are considered [20].

A way of prevent the risks of business failure is the systematic relationship between companies and the market, meeting and even anticipating the customers' needs [21]. Therefore, the marketing performs an important point of articulation between a company and its customers, through the establishment of a systematic interaction between them promoting to companies the perception and understanding of the "voice of the customer" [22]. To this purpose, [22–24] point out some three important aspects: (1) a permanent interaction between the company and the customer to provide exchange of information, namely suggestions from the customer and a validation-built step by step between the designer and the customer, concerning the phases of products or services design; (2) in cases of products or services customization, it is crucial to considerate that the solution was obtained after experiments validated by the customer; (3) the existence of suppliers involvement with the company to ensure the sustainability the design of new products or services from the beginning.

In SIPSNP projects it is very important to consider, in strategic terms, if innovation requires the marketing function, because the role played by it in the whole new products or services design also depends on the level of innovation required, in order to obtain a positive trade-off regarding marketing/quality/cost/time [25].

From a general perspective, obtained from the literature review on the most relevant factors that comprise a strategic vision in SIPSNP environment described above, is presented in Figure 2 with a thereof summary of SIPSNP.

**Figure 2.** Systemic and strategic environment. Framework Approach (authors' own elaboration).

*2.2. Organizational Parameters*

From literature review it was found that several parameters have influence in firms that develop new products with sustainable innovative processes, in a structural way, and that they reported to the corporate culture; and the ones of a conjunctural order were derivate from management principles. It was also enlighten by the existing literature that each type of parameters is associated to specific factors, which will be covered in the following two sections.

2.2.1. Corporate Culture

In strategic management, the organizational parameters are considered part of the corporate culture. And one of the most relevant is the ability to function in the development of sustainable innovative projects with cross-functional teams perfectly connected in a systematic way. Therefore, many authors advise multidisciplinary, multifunctional and/or cross-functional organization type [1,26–29]. Such connections pass through the formation of collaborative teams, which should include at least employees

of the organization, suppliers and customers. So, it is crucial to have a reliable information flow that ensure visibility and transparency in connecting people, processes and technologies.

The organizational strategy of working in multidisciplinary integrated teams (cross-functional) is increasingly suitable to companies that develop new products and services, due to markets globalization, existing together in partnerships and collaborative alliances with inter-organizational information sharing skills and sustainable innovation [30]. That is, a whole innovation capacity, radical and incremental, in organizations that work on network and that encompass collaboration with customers and suppliers [31].

Another way to characterize the sustainable innovative processes concerns to open innovation, in which resources move easily at the border or interface company/market [32]. Whenever open innovation must be shared as a partnership or strategic alliance, it assumes the designation of co-innovation [33]. This shared innovation, benefits the value chain to the customer, called a win-win relationship, and is of a major importance for companies to create value in the market. According to [34], with co-innovation different internal and external sources are integrated into a platform in order to turn the company more competitive and able to satisfy the customer's voice that means the existence of co-creation, co-design and joint development [35].

It follows that the corporate culture should incorporate another common inter-organizational factor: the competitiveness. That is, to incorporate in the company a competitive spirit associated with the effectiveness and success of sustainable new products and services available to the market [36].

### 2.2.2. Management Principles

Regarding the principles of management, the main organizational parameters that allow companies to respond to the market situations, are the following: compliance with legislation of the product inherent to each of the specific markets regarding new products or services development, manufacturing and commercialization [37]; product standardization that permits conformity with international rules and internal flexibility, facilitating modularization processes [38,39]; certification [39]; the agility of performance [40], connected with lean thinking [41,42] in the search for maximum efficiency, effectiveness and productivity [43].

In order to embody the paradigm of optimal productivity is needed to combine lean practices with flexibility and agility, especially when companies need to manufacture various types of products simultaneously [44,45]. Concluded about the importance to associate lean and agile concepts, so they proposed the term "leagility" aiming to integrate them in the paradigm of Supply Chain Management (SCM) in response to market's needs. Likewise, the terms flexibility and proactive flexibility were combined into the term "adaptability" according to [46]. On Figure 3, it's summarized the relevant factors of SIPSNP integrated at the organizational level.

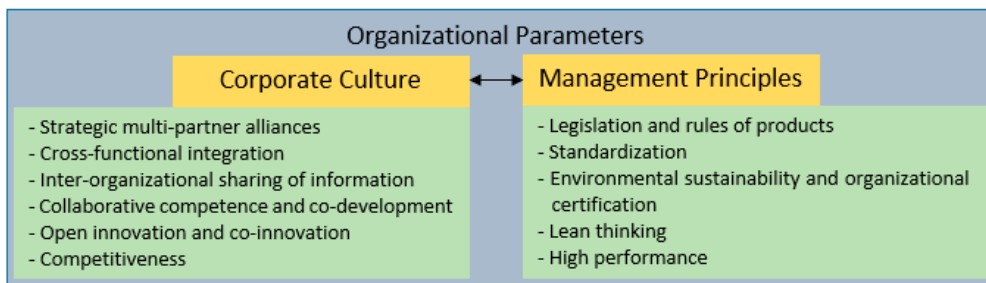

**Figure 3.** Organizational parameters. Framework Approach (authors' own elaboration).

### 2.3. Process Variables

According to literature review, are considered as relevant the following process variables: undertaking an idea of the product or service through a process of innovation management [47];

the organization and management of the project [19]; the quality and control of a project [22] and the engineering and technological capabilities [48,49].

Sustainable innovation management is a methodical process of new ideas generation that allows companies to create value in a proactive way [47], and that can be done in so many ways, therefore that's why each innovation process is unique [50]. From product and service perspective, innovation in a sustainable way to prevent and minimize risks of failure, is therefore a process of creation something new (a product, a service, a process, etc.) yet unknown by the market, and that is due mainly to creative capacity and the technologies available to undertake it. So, it is not a casual situation, but an overall process extending over time [51]. The generation of ideas is a fundamental part of an innovation process that can be convergent or divergent: it is convergent when the idea is the result of a systematic collective process based on trial and error; and it is divergent when a "flash of genius" of some bright and creative collaborator occurs. According to [52], an innovative value chain consists of three main phases: generation of ideas; sorting of ideas and their development; and its dissemination through organization to the market. If it is decided to undertake an innovative idea into a new product or service, the next step will be the project management of SIPSNP.

Many authors as [29,53], present classic models that are examples of "organizational architectures" that group, compose and arrange sub-teams, their inter-relationships and hierarchies. Information flows and "architecture of processes" allow to properly delegate the work to be carried out through the hierarchical levels of companies, as well and the respective flow-related information those levels, aiming to achieve projects goals [20]. Present a model regarding iterative project management, or spiral model, in which flexible changing of work specifications are possible, avoiding the need of restart the whole project from the beginning, modifying only the necessary steps when market changes occur. There are other kind of proposed managing models for SIPSNP projects that are more flexible and agile, widely used by companies worldwide and scientifically known, like simultaneous or concurrent engineering [54–57], and Stage–Gate® [58–60].

Another process variable to have in account in SIPSNP projects is the quality of products and services which are inherent to them [60]. And Taguchi (1986) method is a very important tool used by managers, because of its importance to achieve sufficiently robust outputs with high quality levels [25].

SIPSNP implementation also implies to know and to control engineering and technological capabilities [48,49,61]. Conclude about the importance of prototyping whenever it is needed, and since engineering, technology, quality and reliability are very important issues in those kind of processes, teamwork and collaboration across all hierarchical levels, are crucial for its success. Figure 4 illustrates the most relevant process variables approached by literature review.

**SIPSNP Process Variables**

- Innovation management (new product ideas)
- Project management
- Products quality assurance
- Engineering and technologies available
- Problems and innovative solutions

**Figure 4.** Operational level variables. Framework Approach (authors' own elaboration).

### 2.4. Problems and Innovative Solutions

Markets are increasingly demanding for sustainable and innovative products and services, additionally more information is required by customers, about the environmental impact of products and services provided by companies. Modern management must use sophisticated tools to meet such expectations, so it can be possible to improve monitoring processes of products and services impact, in order to understand how they can be made more sustainable. Regarding products and services lifecycle, which impact is not caused only by the industrial processes or even the usage of

products and services, but also by natural methods of extraction and exploitation of raw materials and others, the transport and storage processes, etc. Therefore, a key factor for a successful sustainability management concerns to the availability and sharing of relevant data and knowledge that must be wisely shared and used along all logistic chain regarding to developed projects [62].

The tools and methodologies available to serve problem solving regarding SIPSNP, are one of the most important issues of a project [63,64]. Conducted a survey of about three dozen tools and techniques obtained through an extensive literature review and realization of several case studies on Taiwanese companies, as well as interviews with experts in the field. Based on [63,64] work, a sample of the most important tools for SIPSNP projects is presented in Table 1.

**Table 1.** Systemic and Strategic Environment in Sustainability of Innovative Processes that Support New Products (SIPSNP) tools and methodologies (authors' own elaboration).

| Survey of Tools to Support SIPSNP | |
| --- | --- |
| **Grouping** | **Tools and Methodologies** |
| Creative and Innovative Solutions | TRIZ; DOE; DFX; Pugh analysis; Creative Design; Axiomatic Design |
| Focus on Quality Function | QFD (e.g.: Kano Model; Ishikawa diagram; DFMEA; Pareto law) |
| Focus on Precision Manufacturing | DFSS (DMAIC cycle and it's variants) |
| Focus on Involvement of Suppliers | SDI |
| Design Support | Robust Design; **Modular** Design; CE |
| Decision Support | AHP; CBR; DEA; Delphi Panel; Fuzzy logics; Neuronal Networks |

Acronyms: TRIZ (Theory of Inventive Problem Solving); DOE (Design of Experiments); DFX (Design for Excellence); QFD (Quality Function Development); DFMEA (Design Failure Model and Effect Analysis); DFSS (Design for Six Sigma); DMAIC (Define-Measure-Analyze-Improve-Control); CE (Concurrent Engineering); AHP (Analytical Hierarchy Process); CBR (Case Based Reasoning); DEA (Data Envelopment Analysis).

Due to the fact that in the literature was found a general confusion about the application of the terms "tools" and "methodologies", for this paper they were grouped putting its focus on their use. Therefore, they were grated as follows: "Creative and Innovative Solutions"; "Quality Function"; "Precision Manufacturing"; "Involvement of Suppliers"; "Design Support" and "Decision Support". Also there was considerate that if similar problems occur, is not guaranteed that they will have similar solutions, because markets are dynamic, and along with that uncertainties can emerge [65]. Uncertainly can be reduced by if the company has an adequate portfolio of problems and respective solutions, and for that the methodology "Case-Based Reasoning" (CBR) can be extremely useful [66]. When a problem is new, there won't be any solutions to solve it obtained from the above mentioned portfolio, so it is necessary to use any of the available methodologies and presented in Table 1. But, if there are several solutions available in such portfolio, it is necessary to determine a ranking of solutions in order to adopt the more suitable one to the existent problem. And for that, one of the most commonly used method is the Analytical Hierarchy Process (AHP). According to [67,68], AHP is very useful to rank the various possible alternatives to support the decision making process. The different ways to achieve a solution to an existing problem, regarding SIPSNP projects, is illustrated in Figure 5.

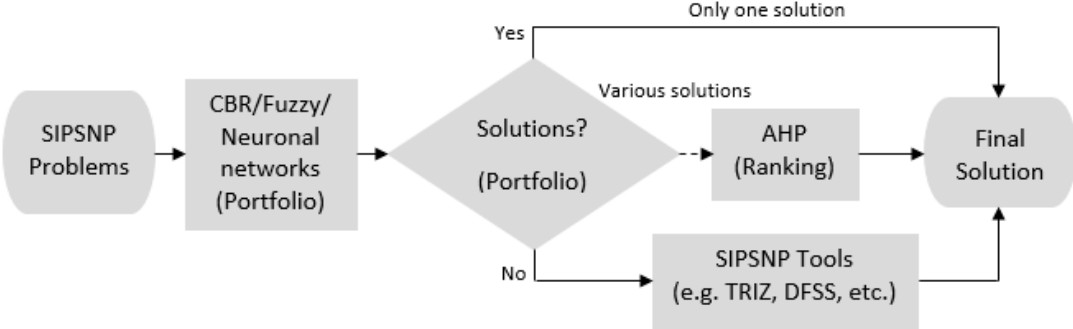

**Figure 5.** Problems and innovative solutions. Framework Approach (authors' own elaboration).

## 3. Research Methodology

As already mentioned, the objective of this investigation is to build a holistic model, which can become a support tool for the sustainable development of innovative processes for creating products and/or services (both incremental and radical), that is, a model that can serve as a roadmap for companies working in this area.

The literature review resulted in a survey, as exhaustive as possible, in a deductive way of the current state of the art and of the set of existing or proposed models for relevant but partial issues of this theme, as long as its methods and tools. The theoretical investigation was conducted in an exploratory way through a deductive-inductive strategy, given that the topic in question adapts to strategic management issues, whose problems and their solutions always involve so many aspects and perspectives, which vary with the changes that constantly occur in the markets that are currently globalized, hampering decision-making processes. It is intended that the main added value of this investigation is evidenced by the difference in the level of science in question between the situation of a possible pulverization of partial models and the construction of the holistic, comprehensive and integrated model, which is intended to be achieved with the work developed.

The proposed model was initially conceptual, as it was obtained thanks to the inductive jump performed after deduction in a qualitative type investigation, having been empirically tested at a later stage, with its external empirical validation through a set of case studies on sustainability innovative processes for creating products and/or services in a national and international industrial environment (in case studies involving projects with partners worldwide). The purpose of this validation was not to generate theory, but only to test, validate and improve it. And so, the conceptual model obtained became functional, because it was shown that it actually works in an industrial environment.

Therefore, it was decided from the beginning on a deductive-inductive structure research, through literature review that it was possible to achieve the conceptual model SIFSNPIP, and there was the need for its empirical validation. Then, for the validation of the model case studies in industrial environment were performed, since most issues to validate were questions of "how" and "why" type, in their qualitative and explanatory variant, as recommended by [69]. In accordance with this work goals, the research was generally regarded as descriptive, due to the fact that it aims to accurately describe the phenomena of reality studied and hence did not require the use of techniques and statistical methods. The methodologies used to validate models regarding exact sciences are often quantitative, while approaching social sciences they are often qualitative, given its high flexibility [69,70].

In the real meaning of the case studies, the case's target to be analyzed is called "unit of analysis" [71]. According to this definition, the units of analysis regarding this investigation are composed of innovative products and services, whose purpose was to test the proposed conceptual model. Since the case studies were performed though interviews complemented with guided tours of companies' manufacturing facilities (that were only possible to perform in Portugal), an interview script (protocol) was elaborated, according to the guidelines pointed out by [71]: interviews (recorded or not) at the place of analysis; telephone conversations; mail contacts; collection of written documents

or computer data; collection of information from "key informants" (only one or a panel), that should be trustworthy people with the right technical and scientific knowledge, from the inside of the organization. Still according to [71], an important aspect to be agreed by both parts in each case study, is the confidentiality of data or information collected and the hypothesis of firm choose to remain anonymous. A final aspect pointed out by this author, refers to the importance of obtaining a formal authorization from the organization boards and provide to their representatives, as well as the "key informants", to review the material provided.

The criterion used in choosing the case studies presented in this article was based on the fact that its scope covers the objectives stipulated in the investigation. In other words, verify and validate all aspects inherent to the proposed conceptual model for the four product/service/incremental/radical combinations, as much as possible, in the perspectives of the national and international industry. The validation of the proposed model through case studies aims to assess its functionality in a business environment and to verify the need of its improvement, so that it can constitute, in the best possible way, a roadmap to be followed by of companies wishing to create new products and/or services in a sustainable manner in the business markets in which they operate. Figure 6 illustrates the key stages involving the research methodology followed in this paper.

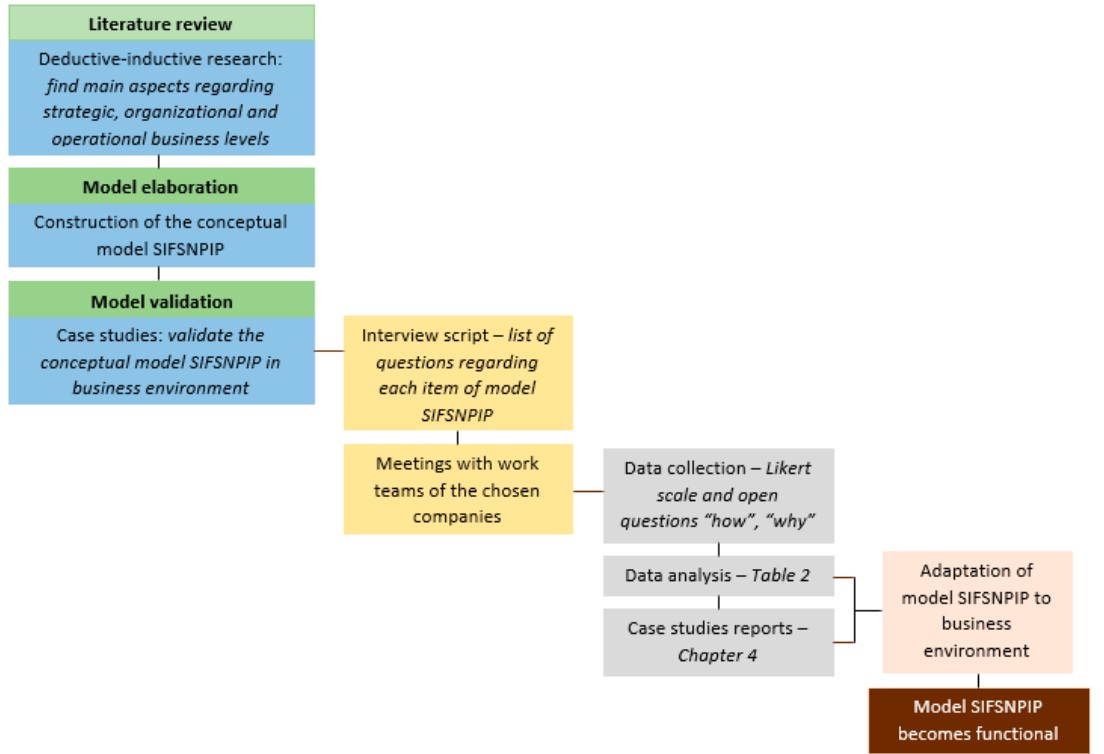

**Figure 6.** Key stages involving the research methodology (authors' own elaboration).

## 4. Case Studies

Four case studies were preformed, regarding products and services, which proved to be sufficient to validate the various parts of the developed model developed, based on literature review, and according to a specific protocol adopted, described by [72]. First, it was validated in the business field the proposed conceptual SIFSNPIP and, secondly, its usefulness was evidenced by demonstrating that it can successfully applied in the assessment of companies that design and develop sustainable new products, allowing to punctuate the evolutionary state of all their strategic, organizational and operational aspects and also its range of innovative products to market.

The ccompanies that collaborated to carry out the presented case studies, were the international business group Instituto de Soldadura e Qualidade (ISQ) and a Portuguese company of metalworking industry, which preferred to remain anonymous.

Of the four explanatory cases studied in industrial environment, two were related to products and were called by "HVAC" and "WJ-LASER" while the other two, were related to services and called by "NaturalHy" and "Brazing".

The case studies were carried out through meetings with work teams of the companies analyzed, belonging to the three levels of decision making: strategic; organizational and operational. For this, three meetings were carried out for each one of the four cases, with a total contribution of 12 work teams. To pursue this end, an interview script was elaborated, composed by questions inherent to each item covered in the model, which was made in accordance with [71,72].

The constitutive meetings of the case studies were aimed at obtaining answers to the questions in the interview script, in order to verify the extent to which all aspects evidenced in the model were performed by the company. The cases are described in Sections 4.1–4.4. Results regarding scores related to the degree of accomplishment by firms of each item of the model, using the Likert scale presented below, are presented in Table 2. Then, complementary open questions of type "how" and "why", made to better understand the extent to which the answers fit this scale to explain the scores obtained, also helped to present the description of the cases and to perform de discussion of the results presented in Section 4.5.

In order to define the scores attributed to each item of the model, the following Likert scale was used:

0. Nothing necessary is accomplished
1. It is performed below the necessary minimum
2. The minimum necessary is accomplished
3. The essentials are performed above the necessary minimum
4. Everything necessary is accomplished

The interview script open questions were designed to address the following aspects in a systemic way:

- Level of achievement of each item in the model;
- Interpretation of the results obtained;
- Maintenance of the model in the short and medium term;
- Elaboration of the conclusions that allowed to confirm (or even improve) the model.

Figure 7 shows that the four case studies were chosen to cover the four possibilities, regarding their application in products and services vs. incremental and radical innovation.

| | Procuct | Service |
|---|---|---|
| Incremental | HVAC | Brazing |
| Radical | WJ-LASER | NaturalHy |

**Figure 7.** Selection criteria for the case studies presented: Application vs. innovation type (authors' own elaboration).

*4.1. HVAC Case*

"HVAC" (Heating, Ventilating and Air Conditioning) case regards to the metalworking industry in which a Small-Medium Enterprise (SME) manufactures a high range of HVAC equipment to be commercialized in the market. This business activity requires a considerable capacity of innovation relating to the manufacture processes of its products, which are mainly: SPIRO® system; heat exchangers; silencers; air handling units (AHUs); rectangular, circular and oval ducts - those ones with

Ethylene-Propylene-Diene-Methylene (EPDM) sealing gasket; chilled beams; fan units; chilled water and storage heat tanks grilles and diffusers and CADvent software (for calculus and dimensioning of air duct installations). To guarantee the sustainability of its products in th metalworking industry e market the firm performs continuous improvement of the products, but because mostly all of them contain a huge number of components, as the AHUs. Therefore, the company can use its resources both to manufacturing and improvement processes, working in ROS to promote its customers satisfaction, and trough marketing, expanding itself in the market.

### 4.2. WJ-LASER Case

"WJ-LASER" case regards the use of both cutting processes water jet (WJ) and (light amplification by stimulated emission of radiation (LASER) on several kinds of materials to promote the manufacturing of customized products, that means, products defined by the customers. The use of this cutting technologies requires the tool Creative Design. The customers that requires this kind of technologies from the firm, are mainly from rehabilitation of ancient artifacts (oil paintings, pottery, papyri, etc.), art and advertising industries. Through the adjustment of cutting parameters in non-metal materials, this two technologies can be used on waste removal of high precision, without damaging the object material. When the firm uses this technologies in the cutting sector, due to the extreme flexibility of its equipment, the innovation level of the products obtained is radical, because all kind of geometries can be designed with high accuracy. Water jet and laser cutting technologies are suitable for small batches or even single parts, but not for mass production.

### 4.3. NaturalHy Case

NaturalHy case regards to a radical service provided by an international business group that focuses strongly on Research & Development (R&D).

The denomination "NaturalHy" defines a project that was recently concluded on which the group participated as part of the executive/steering committee. The project goal is the distribution and use of natural gas with hydrogen addition, so the chemical mix can be used across all Europe, with high level of safety and environmental sustainability through infrastructures designed and built for its distribution. The radical innovation of this case is inherent to the addition of hydrogen to natural gas with a combustion reaction, generating gaseous chemical reaction products with very low amount of carbon dioxide. The project also involved the building of pipelines and storage tanks to distribute and store this new gaseous product, and the distribution network can be used by both domestic and industrial fields, and beyond that, the project also covers permanent monitoring and tracing processes, to ensure its sustainability. The development of this project occurred between years 2004 to 2009, with the collaboration of over 38 business partners, involving a huge investment dimension. The amount invested in the project was about EUR 11 million (granted by the European Commission), having exceeded a profit of about EUR 17 million. Still during the period above mentioned, the project expanded to the Middle East, having participation in the building and operation of the Research Centre of the Petroleum Institute in Abu Dhabi laboratories. This case study was unique to the model purposed in this paper, because allowed it to pass to a functional level with the introduction of exportation policies as a factor not pointed out by literature review on strategic management, as one of the strategic inherent classical issues normally approached.

### 4.4. Brazing Case

Brazing case regards to the use of brazing technology in polymeric materials without lead alloys, as a service with incremental innovation, since this technology was already used worldwide in metal materials. Working with alternative chemical alloys rather than lead, in brazing of polymeric materials (that are the material basis of electronic circuit boards) allows to guarantee health and environmental sustainability. The company offers a testing service using this technology to serve projects regarding the manufacture of electronic and electrical components for several kinds of industries, namely

audio visual, aerospatiale, appliances and electronics firms and business groups, in partnership with: Research & Development (R&D) institutions; airlines; governmental agencies and armed forces. From these business partners are highlighted: Crane-NSWC; American Air Force; Boeing; BAESystems; ITB Inc.; NASA; Texas Instruments; Northrop Grummam and Portuguese Association of Electrical and Electronics Industries (PAEEI). A disadvantage of using this technology is the difficulty on welding in polymeric materials using elements with high melting points, like tin to work with this type of circuits, and for this reason, it is crucial to conduct a high number of tests. When the risk of failure of an electronic board leads to catastrophic results, in the case of aircraft and military armament, lead alloys can be used again, so experiments are extremely important to make a decision about which alloy material can be used with a minimum risk of failure. Since testing is one the most important phase of high risk projects, DFSS, DOE, DFX, among others, are the support methodologies and tools to test performing with the highest accuracy possible.

*4.5. Discussion of Results*

In "HVAC" case it was found that the "systemic and strategic environment" level almost factor scores were of 1 and 2. This is an acceptable fact, because the firm works in ROS. In the factor levels of both "corporate culture" and "management principles", scores incidence occurred were of 1 and 3.

This was an expectable fact, because firm works with compliance by the rules and obligations to the market. Regarding "process variables" level, the scores were almost all between 1 and 2, due to the fact that the firm works with the same range of products aiming to make the best use of its resources and with a specific technological and engineering know-how. And from the panel of tools available in Table 1, the use of modular and tolerance design reached the score 4, because all production is composed of modular products, in which assembling processes need to obey to specific tolerances.

In "WJ-LASER" case, when rating the "systemic and strategic environment" level, almost factor scores were in 4, but not the one regarded to ROS vs. BOS that correspond to score 2, and the one regarded outsourcing with score 0. In the factor levels of both "corporate culture" and "management principles", scores incidence occurred were on 4. The factors regarding involvement of suppliers and customers scored in 2 or 3 respectively, because concerns to a radical innovation. In level "process variables", all scores obtained were equal to 4, because it is a project that must highly obey to all that issues to be extremely profitable. But relating Stage-gate® projects the score was 0, because this management tool is not applicable on water jet and laser technologies. It was found a large application of creative project methodology combined with modular design. And almost all other methodologies and tools referred in Table 1 were pointy used, except the TRIZ and DFSS, because there were no contradictions to be solved through TRIZ methodology, and DFSS has no application in individual parts or small batches of products concerning this case.

In "NaturalHy" case it was found that the "systemic and strategic environment" level, factors scores were of 4, except the one concerning the ROS, because it is the innovation is radical. This case showed up that exportation issue was not contemplated in the conceptual model so, it was a gap not found in the literature review, that when included on the "systemic and strategic environment" level of the conceptual model, turned it into functional. In the literature review about strategic management was found a single paper on this subject—the work of [73] - regarding business expansion and perspectives. Regarding levels of both "corporate culture" and "management principles" most of its factors were scored with 4, but not the ones concerning offshoring, outsourcing, and the need for low cost solutions. Regarding the level "process variables", its factors were scored between 1 and 4, because it is about a project regarding a huge amount of issues to take in account, and ones are more demanding than others. Regarding methodologies and tools presented on Table 1, the business group it was found that almost all of them were applicate in the project.

In "Brazing" case it was found that the "systemic and strategic environment" level, factors scores were practically all on 4. Regarding levels of both "corporate culture" and "management principles" almost all of them were scored with 4, and the same happened to level "process variables". Regarding

methodologies and tools presented on Table 1, it was found that the company uses almost all of them, sometimes separately and other times combining the ones that are complementary. For example, the business group uses DFSS many times, because the level of accuracy required on the manufacturing of electronic circuit boards is a very accurate process.

All scores that were obtained using the already mentioned Likert scale on the application of all SIFSNPIP items, with the four case study carried out, are presented in Table 2.

**Table 2.** Summary of the scores of the factors measured by SIFSNPIP (authors' own elaboration).

| Levels | Parameters and Variables | Cases | | | |
|---|---|---|---|---|---|
| | | HVAC | WJ-LASER | NaturalHy | Brazing |
| **Systemic and Strategic Environment** | Strategy and innovation policies | 2 | 2 | 2 | 4 |
| | Risk analysis and trade-off evaluation | 1 | 4 | 4 | 4 |
| | Marketing policies; customers' and suppliers' engagement | 1 | 2 | 4 | 4 |
| | Benchmarking capacity | 1 | 4 | 4 | 4 |
| | Globalization policies | 2 | 4 | 4 | 4 |
| | Exploitation policies | —— | —— | 4 | —— |
| **Organizational** (Culture) | Strategic multi-partner alliances | 3 | 4 | 4 | 4 |
| | Cross-functional integration | 2 | 4 | 4 | 4 |
| | Inter-organizational sharing of information | 2 | 3 | 4 | 4 |
| | Collaborative competence and co-development | 3 | 4 | 4 | 4 |
| | Open innovation and co-innovation | 2 | 4 | 4 | 4 |
| | Competitiveness | 2 | 4 | 4 | 4 |
| **Organizational** (Management Principles) | Legislation and rules of product | 3 | 4 | 4 | 4 |
| | Standardization | 3 | 4 | 4 | 4 |
| | Environmental sustainability | 3 | 4 | 4 | 4 |
| | Organizational certification | 3 | 4 | 4 | 4 |
| | Lean thinking | 1 | 4 | 4 | 4 |
| | High performance | 2 | 4 | 4 | 4 |
| **Process Variables** | New product ideas (conception and development) | 2 | 4 | 4 | 4 |
| | Project management | 2 | 3 | 3 | 4 |
| | Products quality assurance | 3 | 4 | 4 | 4 |
| | Engineering and technologies available | 2 | 4 | 4 | 4 |
| | Problems and innovative solutions | 1 | 4 | 4 | 4 |
| **Problems and Innovative Solutions** (Methodologies and Tools) | TRIZ; DOE; DFX; QFD; DFMEA; DFSS; DMAIC; CE; AHP; CBR; DEA; etc. | 1 | 3 | 3 | 3 |

The four cases studies regarding both incremental and radical products and services, carried out on firms and business group's facilities, were found to be enough regarding the measurement of all issues that integrate its levels, because the results obtained fit into the analyzed realities. And conceptual model SIFSNPIP, built based on the literature review, became conceptual after its validation in industrial field through the described cases studies.

## 5. Proposal of a Conceptual/Functional Model

SIFSNPIP functional model was finally obtained, by putting Figures 1–5 altogether along with its interactions, as illustrated on Figure 8.

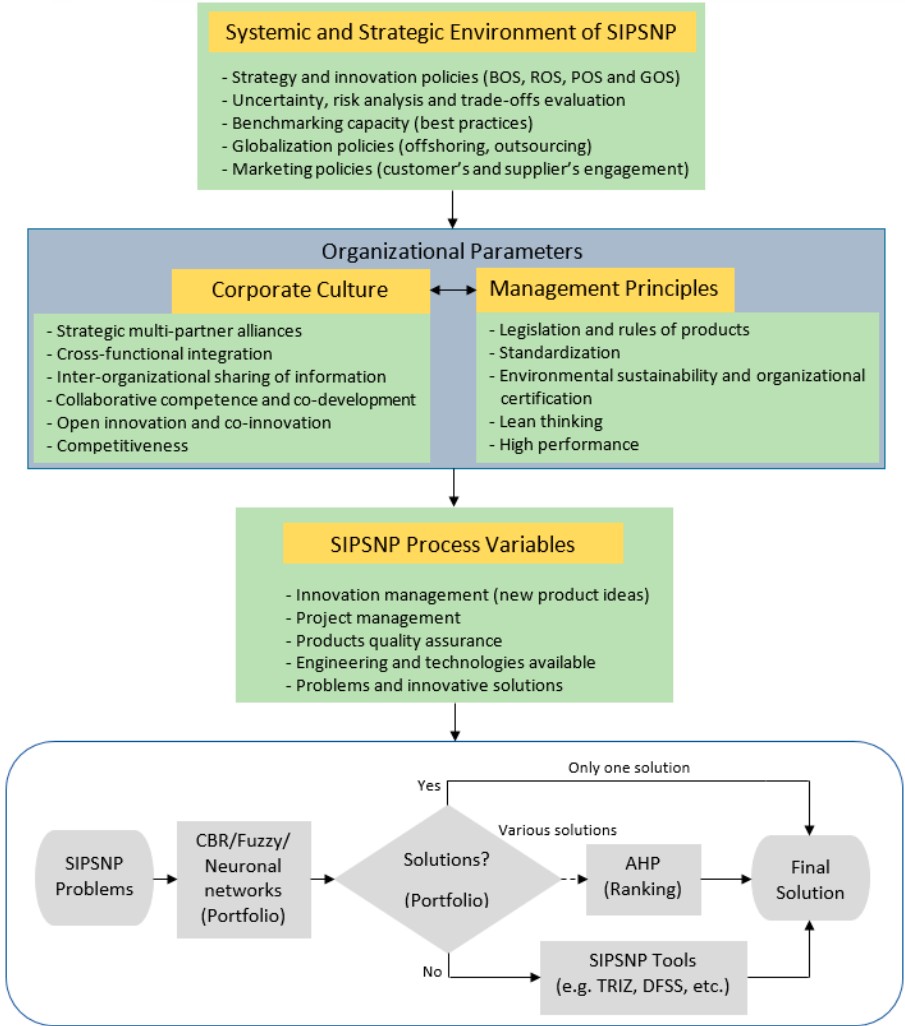

**Figure 8.** Functional model of SIFSNPIP (authors' own elaboration).

It is important to highlight that the item "export policy" integrates the "systemic and strategic environment" level of SIPSNP model, and also that this fact was the differentiator factor between the conceptual and functional model.

## 6. Conclusions

The research on literature review, was based on a deductive-inductive pathway, in order to construct a comprehensive and integrated conceptual model to support Sustainability of New Product Innovation Processes—the SIFSNPIP model. It was empirically validated, in the industrial environment through four explanatory case studies, referring to the implementation of sustainable new products and services, both incremental and disruptive. From the literature review it wasn't detect, until now, any holistic models to support sustainability of new product innovation processes, regarding this phenomenon as a whole appropriate for cases of enterprises or industries models, but only partial approaches. This fact justified the purpose of this investigation, and for all that was exposed in this paper, it was concluded that this goal was successfully achieved. The SIFSNPIP model that was initially conceptual, became functional after its external validation in industrial field through the performance of four case studies, regarding both incremental and radical products and services. So, functional SIFSNPIP model can be used as a diagnostic tool or roadmap for measurement of projects carried out by firms that innovate, design and develop new products with a sustainable way of management.

According to the initial objectives, model SIFSNPIP should allow at least two different uses:

- The first, with a purely scientific nature as an organized menu of solutions to problems that occurred in innovative processes to design new products, using known methodological and instrumental tools;
- The second, with an operational nature, in which the model will work as a diagnostic roadmap for measuring processes, projects, and products, with the purpose of reducing the risks of business failure as much as possible.

It seems clear that would be desirable that model SIFSNPIP could be better tested with a dozen or more case studies, with several business organizations that design new products, both incremental and radical, sustainably. Such applications could have, for organizations, a measurement and improvement of their own processes, in addition to any specific adaptations of the model, as well as data collection that would allowed a statistical treatment of the incidence and influence of the model factors on competitiveness of the national and international innovative industry itself.

**Author Contributions:** Conceptualization, A.S.M.E.D. and A.A.; methodology, A.S.M.E.D., A.A., H.V.G.N. and R.S.; validation, A.S.M.E.D., H.V.G.N. and R.S.; formal analysis, A.S.M.E.D. and A.A.; investigation, A.S.M.E.D., A.A., H.V.G.N. and R.S.; resources, A.S.M.E.D. and A.A.; data curation, A.S.M.E.D. and A.A.; writing—original draft preparation, A.S.M.E.D. and A.A.; writing—review and editing, A.S.M.E.D. and R.S.; visualization, A.S.M.E.D. and R.S.; supervision, A.S.M.E.D. and R.S.; project administration, A.S.M.E.D. and A.A. All authors have read and agreed to the published version of the manuscript.

**Funding:** This research received no external funding.

**Acknowledgments:** The authors of this paper wish to thank the Sustainability Journal for the opportunity to submit this paper to its Special Issue "Innovation Ecosystems: A Sustainability Perspective", a fact that gave inspired motivation to carry out this work. It is also intended to express gratitude to all of the members of the scientific community who, in the past, have written R&D works that served as references for the literature review done in this research work in particular. A "thank you" is sent to the companies that collaborated to carry out the presented case studies (international business group Instituto de Soldadura e Qualidade—ISQ and a Portuguese company of the metalworking industry that preferred to remain anonymous), which contributed to increase the knowledge about the subject in question. Lastly, the authors also wish to acknowledge the Fundação para a Ciência e a Tecnologia (FCT—MCTES) for its contact support that helped to carry out the case studies presented in this paper via the project UIDB/00667/2020 (UNIDEMI).

**Conflicts of Interest:** The authors declare no conflict of interest.

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
