# Peer review of "Proposal of a Holistic Framework to Support Sustainability of New Product Innovation Processes"

_sustainability, doi:10.3390/su12083450_

Round 1

Reviewer 1 Report

This is an interesting paper that includes a detailed coverage of the research study.

Unfortunately the paper is let down by poor grammar. For example the very first sentence in the abstract does not make sense: “Assure the sustainability of innovative processes, whenever the creation of new products is needed, is crucial for the survival of firms, as a source of competitive advantage and a determinant factor of business success”.

There are countless examples throughout the entire paper of poor grammar that need to be rectified. It is suggested that the authors work with an appropriate editing service or a native English speaker to address this matter.

The following figures/tables need correcting as there are various issues to be addressed: Fig. 4, Table 1, Fig. 6.

Further areas to be considered by the authors are as follows:

It would be useful to see the supporting literature references gathered or tabulated for the process variables in Fig. 4.

Table 2 provides summary score data. But how was the source or primary data collected? What data was collected? From whom, and how? How many people were involved in the case studies?

It is not clear how the empirical data was obtained and analysed. It is suggested that further details are included in this area.

Are there any descriptive statistics available for the empirical data? Can a supporting diagram be provided to further illustrate the key stages involved in the research methodology?

Does the research study have any limitations? Also, can these be addressed through future work that should also be identified?

Author Response

Point-by-point response to reviewers’ comments

Reference: Manuscript ID sustainability- 771585

We would like to thank the reviewers for their significant effort and valuable recommendations. We appreciate the feedback that has much value in improving the manuscript. We did revise the article according to received first-round feedback.

Below you can find the original review comments, followed by our responses indicating changes to the manuscript. We have attempted to implement the recommendations to the best of our ability.

The changes have been marked in red color in the revised manuscript rev1.

Manuscript Title: Proposal of a Holistic Framework to Support Sustainability of New Product Innovation Processes

Response to Reviewer 1 Comments

Open Review

English language and style

(x) Extensive editing of English language and style required
( ) Moderate English changes required
( ) English language and style are fine/minor spell check required
( ) I don't feel qualified to judge about the English language and style

Yes

Can be improved

Must be improved

Not applicable

Does the introduction provide sufficient background and include all relevant references?

(x)

( )

( )

( )

Is the research design appropriate?

( )

(x)

( )

( )

Are the methods adequately described?

( )

( )

(x)

( )

Are the results clearly presented?

( )

(x)

( )

( )

Are the conclusions supported by the results?

( )

( )

( )

( )

Comments and Suggestions for Authors

This is an interesting paper that includes a detailed coverage of the research study.

Unfortunately the paper is let down by poor grammar. For example the very first sentence in the abstract does not make sense: “Assure the sustainability of innovative processes, whenever the creation of new products is needed, is crucial for the survival of firms, as a source of competitive advantage and a determinant factor of business success”.

There are countless examples throughout the entire paper of poor grammar that need to be rectified. It is suggested that the authors work with an appropriate editing service or a native English speaker to address this matter.

Thank you very much for this observation. English language or style mistakes have been revised throughout the entire document!

The following figures/tables need correcting as there are various issues to be addressed: Fig. 4, Table 1, Fig. 6.

Further areas to be considered by the authors are as follows:

It would be useful to see the supporting literature references gathered or tabulated for the process variables in Fig. 4.

Thank you very much for the relevant observation! All figures and table 1 of the article were elaborated by the authors based on literature review, and now we are referring to that, in order to comply with the work that we done.

Table 2 provides summary score data. But how was the source or primary data collected? What data was collected? From whom, and how? How many people were involved in the case studies?

It is not clear how the empirical data was obtained and analyzed. It is suggested that further details are included in this area.

We fully agree with this point! In order to elucidate these 2 questions as best as possible, a new paragraph was made in which we tried to explain them as clearly as possible (included between lines 398 and 421 of the article).

Are there any descriptive statistics available for the empirical data?

Once again, we thank you for the valuable observation! No there are not, because we didn’t have funding assistance to carry out more case studies with a larger number of data to proceed with its statistical treatment. With the improvement of the description shown in paragraph included between lines 342 and 346 of the article, and the inclusion of the new paragraph referred before, we hope to have clarified this issue as well.

Can a supporting diagram be provided to further illustrate the key stages involved in the research methodology?

We found your suggestion very helpful and we agree that will enrich the work! To fulfill this issue we have drawn up a supporting diagram (now corresponds to figure 6) that we hope can illustrate the key steps of the research methodology in the best way possible.

Does the research study have any limitations? Also, can these be addressed through future work that should also be identified?

In fact, are very well observed points! Yes, it does have the question of not being able to carry out a large number of case studies, which would allow the realization of statistical data processing. So, we have elaborated a last paragraph in the conclusions, in which we present the limitation of the model - inherent to our impossibility of not having been able to make a sufficient number of cases to allow statistical treatment of the collected data - and we leave our suggestion for future work, hoping to eliminate that limitation in a future work, that it will be important for us to develop in a future article.

Reviewer 2 Report

Dear authors,

I think that this paper is quite interesting. However, I am not expert in the field, then I could evaluate the form and methodology and interest just in general. I think that it is wellwritten and the results are clear. However, I think that you could improve the paper:

  1. Try to explain properly what is the problem and objective of the paper
  2. Try to reduce a bit the number of word, because the paper is very long
  3. Explain what is the foundamentation of the survey properly
  4. Establish clear concluding remarks and implications for academic and practitioners
  5. Reduce the huge number of literature and establish the most relevant one

Author Response

Point-by-point response to reviewers’ comments

Reference: Manuscript ID sustainability- 771585

We would like to thank the reviewers for their significant effort and valuable recommendations. We appreciate the feedback that has much value in improving the manuscript. We did revise the article according to received first-round feedback.

Below you can find the original review comments, followed by our responses indicating changes to the manuscript. We have attempted to implement the recommendations to the best of our ability.

The changes have been marked in red color in the revised manuscript.

Manuscript Title: Proposal of a Holistic Framework to Support Sustainability of New Product Innovation Processes

Response to Reviewer 2 Comments

Open Review

English language and style

( ) Extensive editing of English language and style required
( ) Moderate English changes required
( ) English language and style are fine/minor spell check required
(x) I don't feel qualified to judge about the English language and style

Yes

Can be improved

Must be improved

Not applicable

Does the introduction provide sufficient background and include all relevant references?

(x)

( )

( )

( )

Is the research design appropriate?

( )

(x)

( )

( )

Are the methods adequately described?

(x)

( )

( )

( )

Are the results clearly presented?

(x)

( )

( )

( )

Are the conclusions supported by the results?

( )

(x)

( )

( )

Comments and Suggestions for Authors

Dear authors,

I think that this paper is quite interesting. However, I am not expert in the field, then I could evaluate the form and methodology and interest just in general. I think that it is wellwritten and the results are clear. However, I think that you could improve the paper:

  1. Try to explain properly what is the problem and objective of the paper

Thank you for the valuable alert! Regarding the problem to which the article intends to contribute, as well as its relevance and objective, the explanatory text on this part was changed in the abstract and in the introduction of the article, in order to explain in the clearest possible way what was not previously noticeable.

  1. Try to reduce a bit the number of word, because the paper is very long

Thank you for the observation! So, we rewrote, throughout the article, some explanations that were exposed in a less direct way, in order to make them more succinct. And to better fulfill this objective, we have also eliminated some definitions that are common knowledge in management.

  1. Explain what is the foundamentation of the survey properly

We fully agree with this point! In order to elucidate this question as best as possible, a new paragraph was made in which we tried to explain them as clearly as possible (included between lines 398 and 421 of the article). Completely, we have drawn up a supporting diagram (now corresponds to figure 6) that we hope can illustrate the key steps of the research methodology in the best way possible. We also have elaborated a last paragraph in the conclusions, in which we present the limitation of the model - inherent to our impossibility of not having been able to make a sufficient number of cases to allow statistical treatment of the collected data - and we leave our suggestion for future work, to eliminate that limitation.

  1. Establish clear concluding remarks and implications for academic and practitioners

Thank you for highlighting this topic. Indeed, for helping readers to understand this question as best as possible, a new paragraph was made in which we tried to explain them as clearly as possible (included between lines 606 and 618 of the article).

  1. Reduce the huge number of literature and establish the most relevant one

Once again, we thank you for the observation! In order to better fulfill this point, we not only eliminated references that did not address the central questions of the article (12,13,15,16,19,24,25,27,30,31,35,41,43,45,49,52, 72, 73 and 78 to 80), but we also paid attention to combining this point 5. with that suggested in point 2.